# A comparison of self- and semi-supervised pretraining approaches for risk prediction from chest radiograph images

**Yanru Chen**[1,2]                                           YC4037@COLUMBIA.EDU
**Michael T Lu**[1]                                          MLU@MGH.HARVARD.EDU
**Vineet K Raghu**[1]                                     VRAGHU@MGH.HARVARD.EDU
[1] *Cardiovascular Imaging Research Center, Massachusetts General Hospital  Harvard Medical School*
[2] *Department of Computer Science, Columbia University*

**Editors:** Accepted for publication at MIDL 2023

## Abstract

Deep learning is the state-of-the-art for medical imaging tasks, but requires large, labeled datasets. For risk prediction (e.g., predicting risk of future cancer), large datasets are rare since they require both imaging and long-term follow-up. However, the release of publicly available imaging data with diagnostic labels presents an opportunity for self and semi-supervised approaches to use diagnostic labels to improve label efficiency for risk prediction. Though several studies have compared self-supervised approaches in natural image classification, object detection, and medical image interpretation, there is limited data on which approaches learn robust representations for risk prediction. We present a comparison of semi- and self-supervised learning to predict mortality risk using chest x-ray images. We find that a semi-supervised autoencoder outperforms contrastive and transfer learning in internal and external validation data.

**Keywords:** Deep learning, risk prediction, chest radiographs, self-supervised pretraining

## 1. Introduction

Deep learning [5,24] is the state of the art for medical image segmentation [18] and interpretation [17] tasks ; however, deep learning requires large, labeled datasets, which are rare. One solution is transfer learning, where models are initially trained to classify objects in large natural image data (e.g., ImageNet [11]), and then fine-tuned for the target task. However, ImageNet-based transfer learning may provide limited benefit beyond training models entirely from scratch [30]. In contrast, self-supervised techniques use unlabeled medical images to learn compressed, expressive representations. Due to the feasibility of releasing unlabeled image data, this strategy is promising for medical image analysis [34].

Beyond image interpretation, deep learning can estimate future disease risk [25,28,33,26,31] using datasets with imaging and follow-up. These tasks differ from common pretraining tasks (e.g., object detection, segmentation, etc.) in that there is not necessarily a localizable pathology (object of interest) on the image. For example, frailty, posture, and gestalt suggestive of old age may indicate higher disease risk [41]. In addition, labels are scarce as follow-up information (diagnosis dates, dates of death, etc.) is protected health information. Here, self- and semi-supervised techniques may improve risk prediction by using unlabeled data to improve label efficiency.

Chest radiographs (x-ray or CXR) are an ideal test-bed for risk prediction approaches because they are the most common diagnostic imaging test [32]. Studies have shown that

deep learning models can accurately interpret CXRs [5], and self-supervised techniques [34] improve CXR interpretation beyond transfer learning [37]. These efforts have been supported by the public release of large datasets with radiologists' interpretation of CXRs [17,40,4, 19] (> 1 million total images). For risk prediction, these datasets are "unlabeled" in that there is no follow-up information. It is not yet known whether self- and semi-supervised techniques can leverage such data to improve label efficiency.

In this study, we evaluate self- and semi-supervised pretraining techniques for predicting mortality risk from CXRs. Our specific contributions are:

- We evaluate pretraining strategies using target variables with varying class imbalance.
- We investigate the generalizability of representations in an external validation dataset.
- We assess whether representations can predict disease history and risk factors.
- We explore the impact of image resolution on learned representations

## 2. Related Work

### 2.1. Self-supervised learning approaches

Self-supervised learning specifies a pre-training task on unlabeled datasets to improve performance on a target task with limited labeled data [15]. Most self-supervised learning strategies fall into 1) autoencoder/bottleneck and 2) contrastive learning-based approaches.

#### 2.1.1. Autoencoder-based approaches

Autoencoders have two parts: 1) an "encoder" network that compresses the input image into a low-dimensional encoding and 2) a subsequent "decoder" model that reconstructs the original input image from the encoding [35, 43]. In [43], the authors combine a masked autoencoder (randomly occluded input) with a vision transformer [21] to improve classification of thoracic diseases beyond contrastive learning and ImageNet pretrained models; however, this study only had internal validation. In [35], the authors used a convolutional autoencoder to predict lung nodule malignancy from chest CT images and found a modest improvement over training from scratch. In [14]. the authors show that more generalizable representations are obtained via a variational autoencoder trained on both labeled and unlabeled data with self-ensembling [23]; however, this work lacked external validation.

#### 2.1.2. Contrastive Learning Approaches

In contrastive learning, [12] (Appendix: Contrastive Learning Methodology), a model is trained to create image representations such that "positive pairs" (e.g., images from the same patient) have similar representations and "negative pairs" (e.g., images from different patients) have dissimilar representations [2,39,15,16]. Contrastive-learning improves performance on natural image classification tasks [8, 42]; however, sampling negative pairs has quadratic time complexity. Some use large batch sizes so that the model sees a diversity of negative pairs at each gradient update; however, this is GPU-memory intensive.

Others have proposed using a "memory bank" approach where representations of previously sampled images are stored. Then, representations from the current mini-batch are

"negative pairs" with these stored representations. Thus, the min-batch size is decoupled with the size of the memory bank. One such implementation is momentum contrast or MoCo [15], further described in Appendix: MoCo Methodology

Some approaches avoid using negative pairs entirely (BYOL, DINO, etc.) [6,13]. Instead, a teacher and student network receive augmented views of the same image, and the student must output a "prediction" that matches the teacher's prediction. Typically, the teacher is a moving average of previous student networks.

## 2.2. Comparisons of self-supervised and transfer-based pretraining

One study compared fully-supervised ImageNet pretraining vs. self-supervised pretraining, including SimCLR, SwAV (Swapping Assignments between multiple Views of the same image) [7], DINO [39], and an ensemble approach (Dynamic Visual Meta-Embedding - DVME) on medical image interpretation tasks. For CXR interpretation, no method consistently outperformed; however, self-supervised learning consistently outperformed transfer learning from ImageNet. It is unknown whether these results hold when using in-domain pretraining or in risk prediction tasks. In [2], the authors combined semi-supervised learning with end-to-end contrastive learning as a pretraining method. They compared two image augmentation methods: MICILe (two crops of the same image) vs. SimCLR (rotation, zoom, etc. of the same image). No difference was observed in CXR intepretation tasks. Another work compared [37] a modified MoCo pretrained on CheXpert vs. pretrained on ImageNet. Models were fine-tuned on CheXpert and Shenzhen hospital CXR datasets using 1) an additional linear layer or 2) end-to-end fine-tuning. MoCo models pretrained on CXRs performed better than ImageNet pretrained for all tested label fractions.

These results are corroborated by findings [10] that demonstrate that the pretraining task domain and the quality of the pre-training data are crucial factors for performance. Findings in medical imaging tasks [30] show that simple architectures trained entirely from scratch outperform large models pretrained using ImageNet. To our knowledge, no studies have tested pretraining strategies on risk prediction from medical imaging.

## 3. Methods

### 3.1. Datasets

In our study, we use two imaging datasets for pretraining: CheXpert [17] and NIH CXR-14 [40] (Appendix Figure A.6). CheXpert contains 224,316 CXRs taken at Stanford Hospital between October 2002 and July 2017. NIH CXR-14 contains CXRs taken at the NIH Clinical Center between 1992 and 2015 (108,948 images from 32,717 patients). Only frontal, posterior-anterior CXRs from patients aged 18 to 100 years were included (29419 CXRs from CheXpert and 64628 from NIH CXR-14). Since no follow-up is available, we use age, sex, and acute findings (Appendix I) from the radiologist's read for semi-supervised pretraining. For hyperparameter optimization, CheXpert and NIH CXR-14 were split into training ($N = 74984$) and tuning ($N = 19063$) datasets using a random 80%-20% split by patient. These datasets are used solely for pre-training.

All models were fine-tuned using data (Appendix Figure A.5) from the Prostate, Lung, Colorectal, and Ovarian cancer screening trial (PLCO) [27,29]. PLCO was a randomized

trial where adults 55-74 years were randomized to either a CXR or control (no imaging) with a maximum of 18 years of follow-up. The test dataset includes CXRs from PLCO (40,000 $T_{min}$ - earliest CXR for each participant) and the CXR arm of the National Lung Screening Trial (NLST; 5,414 $T_{min}$ CXRs) [1]. The NLST enrolled adults 55-77 years who had a heavy smoking history. NLST participants had a maximum of 12 years of follow-up. Models were trained to predict all-cause mortality 1 and 12 years after the CXR. Detailed cohort characteristics for all datasets are available in Appendix J.

### 3.2. Pre-training approaches

We compare several pre-training approaches: semi- and self-supervised autoencoder, semi- and self-supervised Patient-Centered Learning of Representations (PCLR), semi- and self-supervised MoCo, transfer learning, and training from scratch.

#### 3.2.1. SEMI-SUPERVISED AUTOENCODER ARCHITECTURE

The Semi-Supervised Autoencoder architecture has three components (Figure 1):

1. An image encoder $f(\cdot)$, which given input image $x_i$, outputs a 512-dimensional encoding $f(x_i) = h_i$. The encoder consists of 4 convolutional blocks, each with a convolutional layer (7x7 filters), batch normalization, ReLU activation, and max-pooling (3x3 window and 2x2 stride) (Figure 1). This architecture was previously shown to outperform transfer learning from ImageNet in medical image interpretation [30].

2. A prediction module, which uses image encodings $h_i$ to classify 14 radiologist findings, (see Section 3.1) sex, and a continuous estimate of age. This encourages representations useful to identify important pathology and demographics.

3. An image decoder $g(\cdot)$, which reconstructs the input image $x_i$ given encoding $h_i$. The decoder has 4 backward basic blocks, all with upsampling, batch norm, ReLU activation, and last layer sigmoid activation (Figure 1).

#### 3.2.2. SEMI-SUPERVISED AUTOENCODER LOSS

The model weights are optimized using a three-part loss function:

1. First, reconstruction mean-squared error (MSE) of predicted vs. original images.

2. Second, a composite cross-entropy loss for multi-label classification (findings and sex) and MSE loss for age.

3. Third, an L1 Regularization on the encodings $h_i$ [3].

The overall loss is given by:

$$Loss_{autoencoder} = \lambda_{recon} * recon_{x,\hat{x}} + \lambda_{reg} * (reg_{h_i} + cls_{h_i}) + \lambda_{norm} * norm \qquad (1)$$

Here, $\lambda_{recon}$, $\lambda_{reg}$, and $\lambda_{norm}$ control the contributions of reconstruction, classification, and normalization to the overall loss. We chose $(1, 20, 0.0001)$ as defaults to scale each equally. We tested the sensitivity of $\lambda$ and found little impact (Hyperparameter Sensitivity Analysis).

The **Self-Supervised Autoencoder** has the same architecture but no prediction module and no cross-entropy loss component.

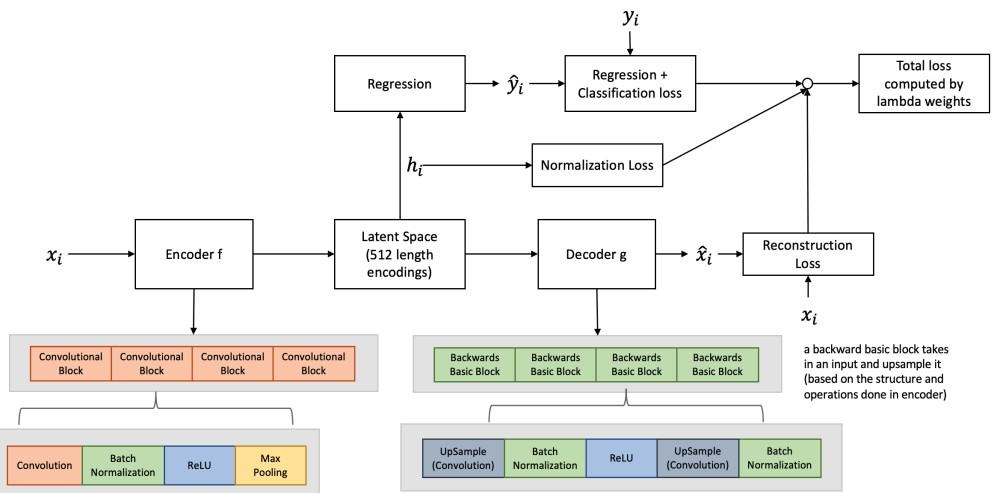

Figure 1: Autoencoder architecture for an input CXR $x_i$ and corresponding label $y_i$.

### 3.2.3. OTHER APPROACHES

**Transfer Learning & Train From Scratch**   We tested transfer learning [38], where the pretraining task was to predict sex, the radiologist findings, and age. We also tested a model that was trained with randomly initialized weights on the PLCO training dataset ("train from scratch"). All approaches use the encoder architecture described above (Section 3.3.1). The transfer learning loss was just the composite cross-entropy and multi-label classification loss (see 3.3.2). A binary cross-entropy loss was used for training from scratch.

**Contrastive Learning**   We used two implementations of contrastive learning: 1) an end-to-end Patient Contrastive Learning of Representations (PCLR) [12] and Momentum Contrast-CXR (MoCo-CXR) [37] A detailed explanation of both approaches are in Appendix: Contrastive Learning Methodology.

### 3.2.4. MODEL OPTIMIZERS AND OPTIMIZATIONS

All models were trained for 100 epochs, batch size 64, using the Adam optimizer [20] with a maximum learning rate of $1e-5$ and the 1-cycle policy [36]. For each strategy, the model with the lowest validation loss during training was chosen for downstream experiments.

### 3.3. Experiments

Experimental steps are outlined in Figure 2. After pretraining, we perform forward propagation on all PLCO and NLST images. The extracted encodings from PLCO participants are divided into training and testing sets (80% - 20% split). The testing set (N = 40,000) is fixed for all experiments, and we randomly sample from the training set to generate 15 datasets for each sample size (40000, 20000, 10000, 5000, 2000, 1000, 500, and 200). Identical training and testing sets were used for all models to ensure fairness. We test the

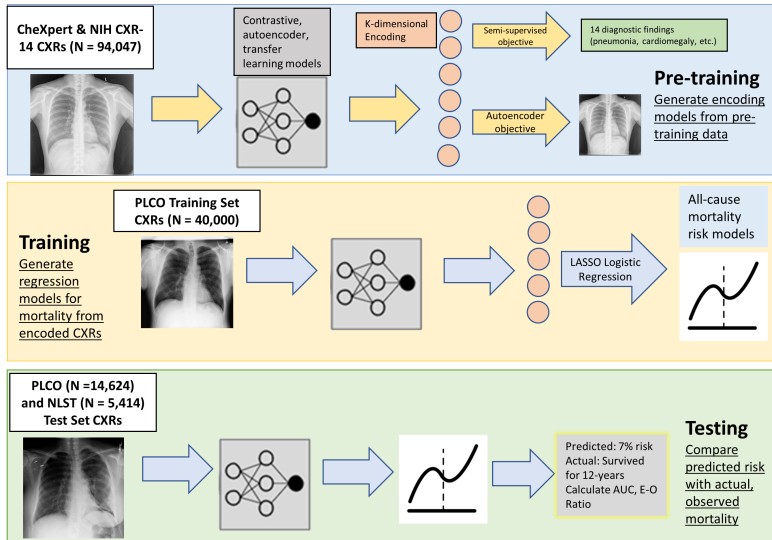

Figure 2: Overview of experimental procedure.

effect of imbalanced outcomes by training on 1-year (event rate 0.456%) and 12-year mortality (14.205%). We separately tested the effect of random weight initializations during pretraining using 5 random initializations for each model (Appendix K).

PLCO encodings were scaled and centered and 0 variance encodings were removed. Testing encodings were transformed using training data statistics. A LASSO logistic regression was trained to predict mortality risk using extracted image encodings. $\lambda$ for LASSO was selected using cross-validation AUC [22]. Pretraining strategies were assessed using discrimination and calibraiton of downstream predictions. We used mean AUC for discrimination and mean E/O Ratio (expected/observed ratio) for calibration across the 15 trials for each training set size and mortality time horizon.

## 4. Results

### 4.1. Effect of pre-training strategy on discriminative accuracy

In Figure 3 and Figure 4, we compared discrimination for 12- and 1-year all-cause mortality. We found that semi-supervision outperformed self-supervised models in external validation for most training set sizes. In particular, the semi-supervised autoencoder and transfer learning from diagnostic labels (with no self-supervision) had highest AUC when the training set was small, while semi-supervised contrastive learning excelled with >5000 samples and for 1-year mortality. Among the fully self-supervised models, contrastive learning (PCLR) performed the closest to semi-supervised methods and even outperformed semi-supervised MoCo in external validation for 12-year mortality. Often, training a model from scratch outperformed self-supervised approaches. For all approaches, the gap between internal and external validation AUC was higher for 12-year mortality than 1-year, potentially due to overfitting. This gap was still pronounced for 1-year mortality for transfer learning, self-supervised PCLR, and training from scratch (> 5% drop in AUC) (Figure 3 and Figure 4).

In a sensitivity analysis, we tested whether the differences between internal and external validation performance could be explained by differences in smoking history. When limiting the internal testing dataset to only individuals meeting eligibility criteria for the external validation dataset, we found that discrimination performance was similar between NLST and PLCO heavy smokers for 1-year mortality (Appendix I), This diffeence was larger for 12-year mortality; however, this suggests that some of the drop in performance in NLST is due to increased smoking intensity, not generalization error. Additional sensitivity analyses showed similar results when changing the image resolution to 128x128 and 320x320 (Appendix L), when varying the random weight initialization (Appendix K), and when changing the downstream classification model (Appendix H).

## 4.2. Effect of pretraining strategy on calibration

We assessed calibration using the E-O Ratio (Appendix Table D.2, Appendix Table D.3, Appendix Table D.4, Appendix Table D.5). In internal validation, we find that all models are well-calibrated for 12-year mortality (except train from scratch which over-predicted risk), and all models over-predicted risk for 1-year mortality. In external validation, the semi-supervised autoencoder and transfer learning were well-calibrated, especially for 1-year mortality. Contrastive learning approaches consistently under-predicted risk for both 12- and 1-year mortality. Self-supervised models followed a similar trend, except contrastive learning alone was well-calibrated in 12-year external validation, whereas the self-supervised autoencoder was well-calibrated for 1-year external validation. In general, semi-supervised models had better calibration in external validation than their self-supervised counterparts.

## 4.3. Comparison of pretraining strategies to predict risk factors

To assess whether some representations were more suited to certain disease processes (e.g., cardiovascular vs. lung disease), we used generated encodings to develop prediction models for prevalent risk factors, disease history, and radiologist findings (Appendix: Intermediate Risk Factors Results). We found that the semi-supervised autoencoder predicted all risk factors better than other pretraining approaches in internal and external validation.

## 5. Discussion

In this study, we tested pretraining approaches to use unlabeled or off-labeled data to predict mortality risk from CXR images. Our major findings were that 1) using off-target, but relevant labels in pre-training is better than self-supervised and no pretraining, 2) self-supervised pretraining is no better than training from scratch in most downstream tasks (Figures 3 and 4), and 3) semi-supervised autoencoder and transfer learning representations were strongly associated with risk factors, suggesting that these representations may be well-suited for more general tasks (Appendix G). We found that autoencoder representations were robust to hyperparameter choices and were generally well-calibrated (Appendix D). In sensitivity analyses, we found that the reduced external validation performance for semi-supervised approaches was due to differences in participant characteristics, not generalization error (Appendix I). Unlike previous studies of medical image pretraining, our study

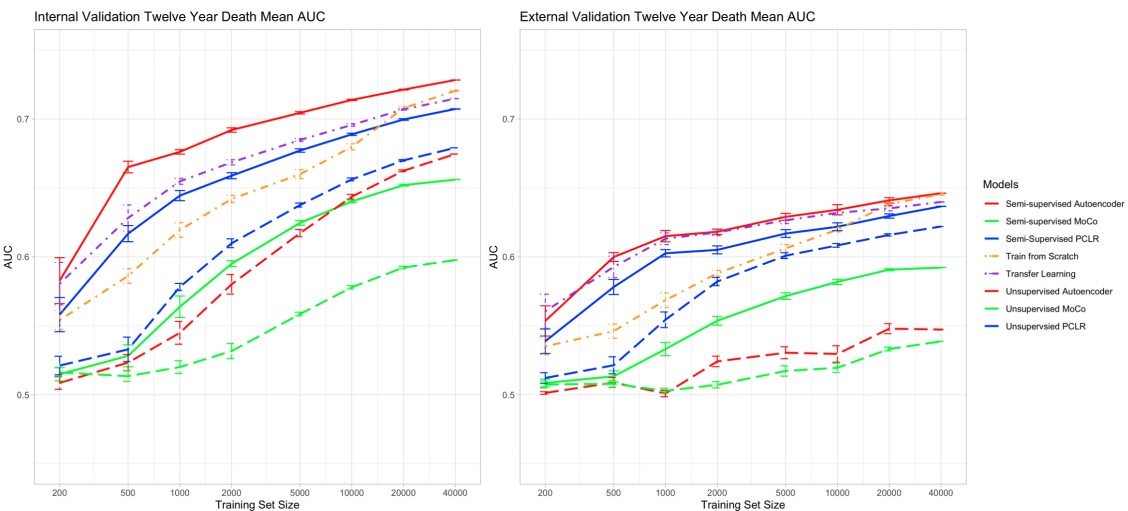

Figure 3: Comparison of approaches to predict 12-year mortality. Training set size (x-axis) vs. mean AUC (y-axis) in PLCO internal (left) and NLST external validation (right) data.

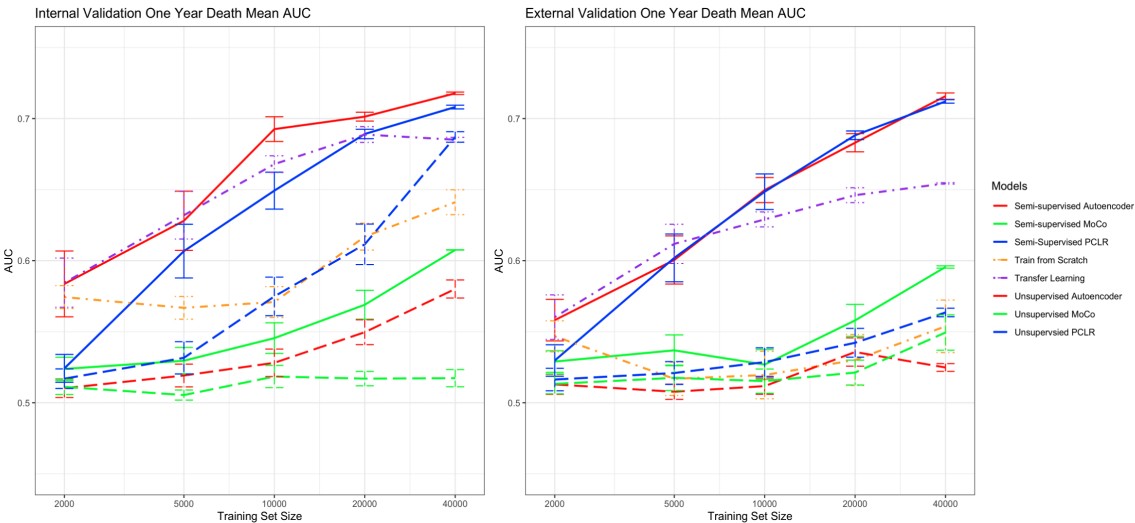

Figure 4: Comparison of approaches to predict 1-year mortality. Training set size (x-axis) vs. mean AUC (y-axis) in PLCO internal (left) and NLST external validation (right) data.

focuses on risk prediction and has external validation data. The semi-supervised autoencoder is publicly available at https://github.com/circ-ml/cxr-encoder. Limitations of this study should be considered. We used one base architecture that showed success in CXR interpretation tasks, and we tested only CXR images; future studies should explore more architectures and other imaging modalities. Our primary outcome was all-cause mortality; however, performance in predicting other intermediate outcomes (heart attack, lung cancer) is unknown. Lastly, our cohorts were predominantly non-Hispanic white individuals.

## Acknowledgments

We would like to thank The National Cancer Institute and the American College of Radiology Imaging Network (ACRIN) for access to trial data, and Stanford University and the National Insitutes of Health for access to the chest x-ray datasets. We also thank the fastai and PyTorch communities for the development of open-source software. The statements contained herein are solely those of the authors and do not represent or imply concurrence or endorsements by any named organizations.

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

## Appendix A. Image Preprocessing

Image preprocessing steps followed our previous work [31]. For PLCO CXRs, we converted original TIF files to PNG with a short-axis dimension of 512 using ImageMagick. A previously developed CNN was used to identify rotated PLCO radiographs and ImageMagick was used to correct these. Similar preprocessing steps were used for NLST, though first CXRs were converted from native DICOM to TIF using the DCMTK toolbox. All input PNG images to the model were resized to 224 x 224 pixels using random cropping along the longer axis. Random data augmentation was used during training, including up to 5 degrees of rotation, up to 20% zoom in/out, and up to 50% brightness and contrast adjustment. No augmentation was used during validation and testing.

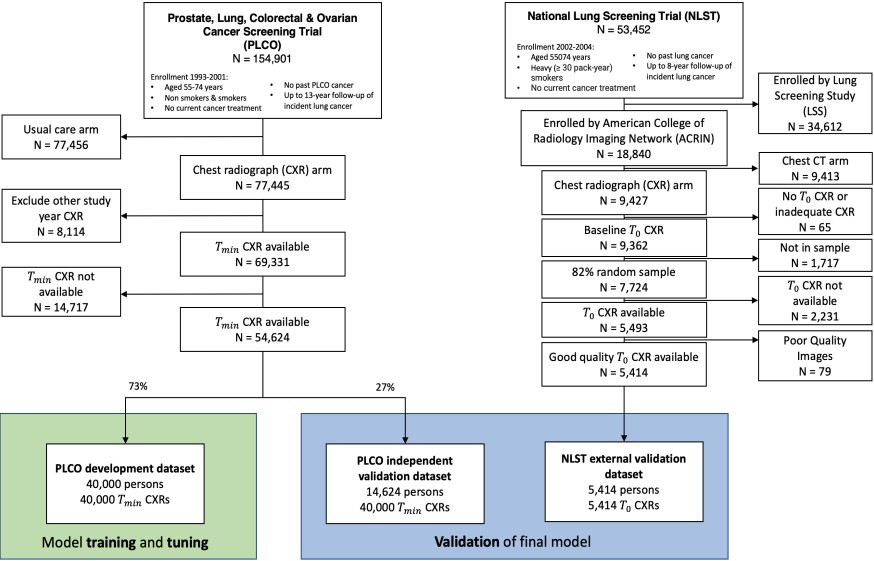

Appendix Figure A.5: CONSORT diagram for PLCO and NLST training and testing datasets. Only the earliest $(T_{min})$ CXR from each patient was used.

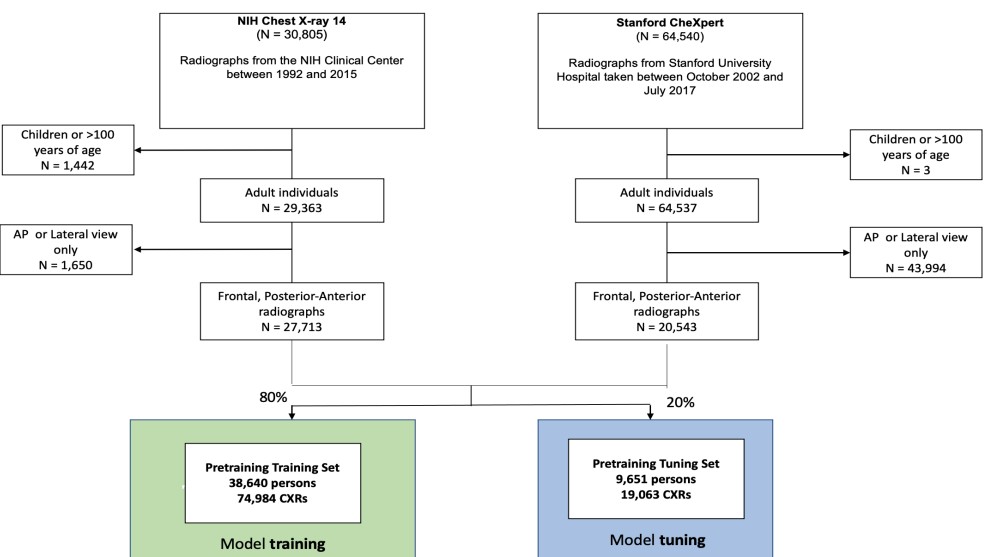

Appendix Figure A.6: CONSORT diagram for pretraining datasets NIH Chest X-ray 14 and CheXpert. Only adult, frontal, and posterior-anterior radiographs are kept.

## Appendix B. End-to-end Contrastive Learning Methodology (PCLR)

We implemented the same data selection method used in PCLR [12] but with a different model architecture. For training, first, a mini-batch of images from unique patients was randomly sampled. To create "positive pairs," a second image from each of these patients was randomly sampled, otherwise, an augmented version of the first image was used. All individual images were randomly transformed as described in Appendix A

The Contrastive Learning approach used an identical image encoder architecture as the Autoencoder-Classifier but trained with different loss functions and input data structure. The major differences in this approach are as follows:

1. A Nonlinear projection head using the ReLU activation function that maintains 512-dimensional encodings and projects image encodings onto a nonlinear space. The results of [8] demonstrate that the incorporation of a nonlinear projection head leads to an improvement in the quality of the representation learned by contrastive learning models. The aim of the projection head is to learn a compact, meaningful and discriminative representation of the input data that can be effectively used in later downstream tasks.

2. A Contrastive loss function is used to train the encoder and projection head. Contrastive loss $cl_{i,j}$ will be low for positive pairs and high for negative pairs, which encourages the model to generate similar encodings (based on cosine similarity) for CXRs from the same patient and dissimilar encodings for CXRs from different patients.

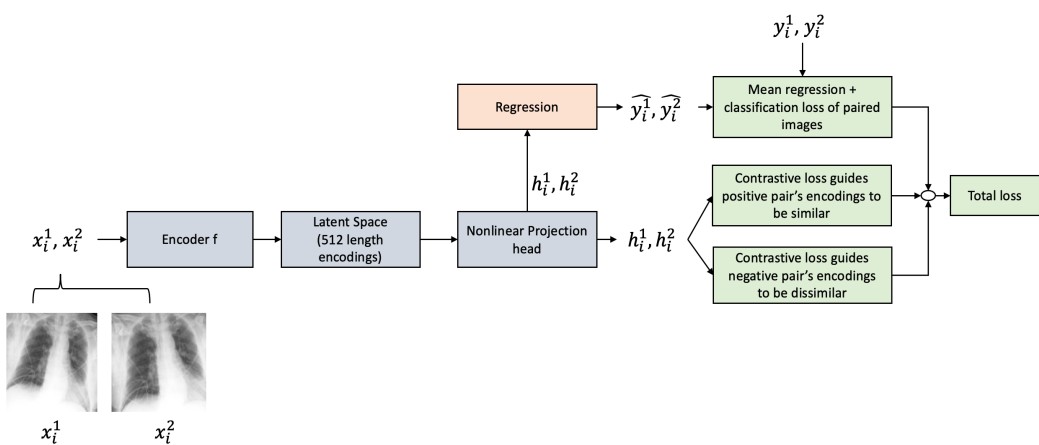

Appendix Figure B.7: Contrastive Learning model architecture for an input x-ray image pair $(x_i^1, x_i^2)$
and corresponding label $(y_i^1, y_i^2)$.

**PCLR Loss**    To make image encodings $p(h_i)$ produced by the projection head $p$ similar for positive pairs and dissimilar for negative pairs, we used the contrastive loss function

introduced in SimCLR [9]. Let $(z_i, z_j)$ be positive pairs, remaining $(z_i, z_k)$ be negative pairs, $\tau$ be the temperature of the similarity function, loss of a positive pair $(z_i, z_j)$ (in total of $N$ pairs of input) using cosine similarity $sim$ as the distance between two encodings can be written as:

$$l_{i,j} = -\log \frac{exp(sim(z_i, z_j)/\tau)}{\sum_{k=1}^{2N} \mathbb{1}_{k \neq i} exp(sim(z_i, z_k)/\tau)} \tag{2}$$

Given a batch of M patients, we will have a total of 2M encoded images and M-positive pairs. Two adjacent encoded images $h_i, h_{i+1}$ will be positive pairs. Thus total loss will be the sum of all losses between positive pairs is:

$$C_{batch} = \frac{1}{2M} \sum_{i=1}^{M} [l_{i,i+1}]_{i \in 2\mathbb{Z}} \tag{3}$$

## Appendix C. MoCo Methodology

We used the MoCo pretraining technique as described in[15] with a batch size of 32 anchor images. A heavy image augmentation technique, similar to the procedure used in SimCLR, is employed as a pretraining task.

The MoCo pre-training procedure is as follows Figure C.8:

1. An image encoder $\theta_q$ used to generate a 512-dimensional image encoding for anchor image (or query, which is the name used in [15]). The structure of this image encoder is the same as that of a complete autoencoder. The image encoding is then projected onto a 128-dimensional vector space using a nonlinear projection head, denoted by $g(\cdot)$.

2. A momentum encoder $\theta_k$ is used to encode the positive and negative pair for the query. The structure of $\theta_k$ is exactly the same as $\theta_q$, but no back-propagation is performed. The weights were updated using $\theta_k \leftarrow m\theta_k + (1 - m)\theta_q$ [15] where $m$ is the momentum hyperparameter which controls the speed of momentum encoder's update. We set $m = 0.999$ as suggested.

3. A first-in-first-out (FIFO) queue, comprised of previous image encodings (keys), enqueues the latest mini-batch's encodings, and dequeues the earliest mini-batch's encodings. Since MoCo is more effective when the dictionary size is large, we used 65,536 for the queue size as suggested by the authors.

4. A contrastive loss function Equation 2 was used to train the overall model with back-propagation performed only on the parameters of $\theta_q$. It uses the query image and its augmented version as a positive pair, which is encoded by $\theta_k$, and the rest of the keys in the queue as negative pairs. The softmax temperature was 0.07.

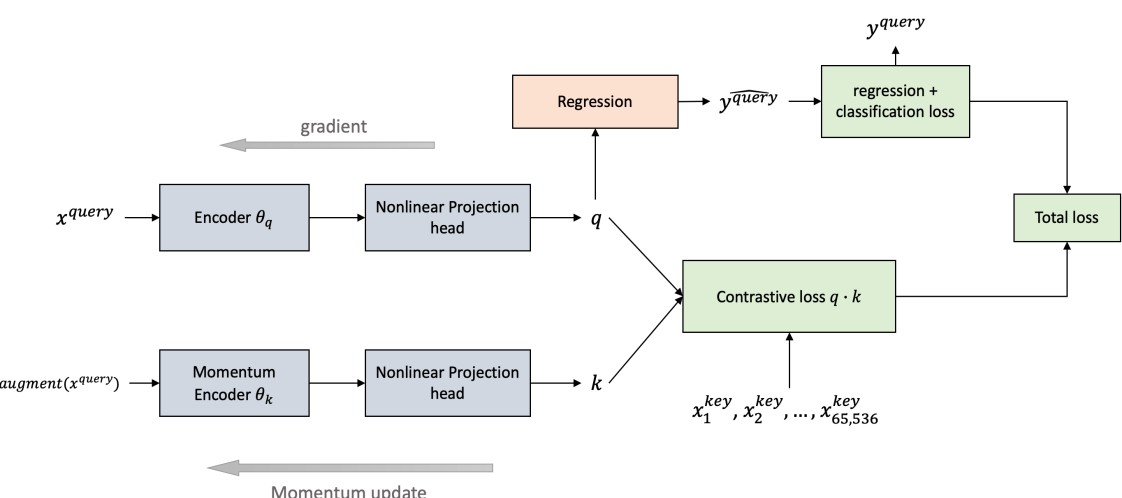

Appendix Figure C.8: Momentum Contrastive Learning model architecture for an input x-ray $x^{query}$, its augmented version as positive pair $augment(x^{query})$, and corresponding label $y^{query}$.

# Appendix D. Calibration of all-cause mortality risk estimates

| Lambda | One-Year Death (PLCO) | | | | Twelve-Year Death (NLST) | | | |
|---|---|---|---|---|---|---|---|---|
| | Amplified Classification Parameter | Amplified Image Reconstruction Parameter | Amplified Normalization Parameter | Original Parameter | Amplified Classification Parameter | Amplified Image Reconstruction Parameter | Amplified Normalization Parameter | Original Parameter |
| 200 | 1.026(0.215) | 1.027(0.212) | 1.021(0.221) | 1.031(0.211) | 0.998(0.535) | 1.104(0.921) | 1.071(0.827) | 1.04(0.513) |
| 500 | 1.054(0.078) | 1.056(0.08) | 1.05(0.08) | 1.047(0.079) | 1.218(0.66) | 0.925(0.584) | 1.098(0.719) | 1.137(0.672) |
| 1000 | 1.014(0.067) | 1.01(0.066) | 1.009(0.07) | 1.008(0.067) | 1.601(0.83) | 1.127(0.844) | 1.285(0.646) | 1.166(0.65) |
| 2000 | 1.021(0.057) | 1.024(0.055) | 1.026(0.055) | 1.023(0.057) | 1.857(0.754) | 1.229(0.685) | 1.127(0.622) | 1.483(0.696) |
| 5000 | 1.025(0.032) | 1.024(0.032) | 1.024(0.03) | 1.023(0.031) | 1.433(0.42) | 0.99(0.585) | 0.631(0.239) | 0.835(0.558) |
| 10000 | 1.143(0.124) | 1.136(0.116) | 1.168(0.124) | 1.262(0.311) | 1.133(0.117) | 1.03(0.013) | 1.031(0.018) | 1.034(0.021) |
| 20000 | 1.227(0.085) | 1.221(0.086) | 1.224(0.103) | 1.162(0.106) | 1.223(0.086) | 1.031(0.015) | 1.033(0.017) | 1.036(0.011) |
| 40000 | 1.212(0.0) | 1.206(0.0) | 1.209(0.003) | 1.187(0.102) | 1.21(0.001) | 1.029(0.0) | 1.031(0.0) | 1.031(0.0) |

Appendix Table D.1: Calibration Results for Hyperparameter Sensitivity Analysis

| PLCO | One-year death | | | | | Twelve-Year Death | | | | |
|---|---|---|---|---|---|---|---|---|---|---|
| | Semi-Supervised Autoencoder | Semi-supervised PCLR | Transfer Learning | From Scratch | Semi-supervised MoCo | Semi-Supervised Autoencoder | Semi-supervised PCLR | Transfer Learning | From Scratch | Semi-supervised MoCo |
| 200 | N/A | N/A | N/A | N/A | N/A | 1.024(0.21) | 1.024(0.206) | 1.018(0.116) | 1.17(0.831) | 1.027(0.211) |
| 500 | N/A | N/A | N/A | N/A | N/A | 1.052(0.081) | 1.052(0.079) | 1.106(0.108) | 1.142(0.179) | 1.057(0.08) |
| 1000 | N/A | N/A | N/A | N/A | N/A | 1.008(0.061) | 1.003(0.066) | 1.015(0.06) | 1.141(0.115) | 1.006(0.067) |
| 2000 | 1.116(0.417) | 1.114(0.412) | 1.223(0.316) | 1.881(1.59) | 1.102(0.435) | 1.022(0.057) | 1.017(0.055) | 1.039(0.036) | 1.108(0.096) | 1.016(0.058) |
| 5000 | 1.222(0.249) | 1.201(0.249) | 1.23(0.259) | 1.226(0.445) | 1.207(0.252) | 1.019(0.029) | 1.021(0.033) | 1.034(0.025) | 1.078(0.054) | 1.022(0.032) |
| 10000 | 1.143(0.124) | 1.136(0.116) | 1.168(0.124) | 1.262(0.311) | 1.133(0.117) | 1.03(0.013) | 1.031(0.018) | 1.034(0.021) | 1.115(0.031) | 1.031(0.016) |
| 20000 | 1.227(0.085) | 1.221(0.086) | 1.224(0.103) | 1.162(0.106) | 1.223(0.086) | 1.031(0.015) | 1.033(0.017) | 1.036(0.011) | 1.112(0.026) | 1.032(0.017) |
| 40000 | 1.212(0.0) | 1.206(0.0) | 1.209(0.003) | 1.187(0.102) | 1.21(0.001) | 1.029(0.0) | 1.031(0.0) | 1.031(0.0) | 1.081(0.032) | 1.031(0.0) |

Appendix Table D.2: Calibration of semi-supervised and supervised models in the PLCO dataset. Results are not available for some smaller training sets due to insufficient events.

| NLST | One-year death | | | | | Twelve-Year Death | | | | |
|---|---|---|---|---|---|---|---|---|---|---|
| | Semi-Supervised Autoencoder | Semi-supervised PCLR | Transfer Learning | From Scratch | Semi-supervised MoCo | Semi-Supervised Autoencoder | Semi-supervised PCLR | Transfer Learning | From Scratch | Semi-supervised MoCo |
| 200 | N/A | N/A | N/A | N/A | N/A | 0.997(0.452) | 0.864(0.224) | 0.881(0.287) | 1.435(0.654) | 0.796(0.157) |
| 500 | N/A | N/A | N/A | N/A | N/A | 1.005(0.393) | 0.907(0.096) | 1.031(0.356) | 1.163(0.293) | 0.824(0.057) |
| 1000 | N/A | N/A | N/A | N/A | N/A | 1.01(0.223) | 0.86(0.106) | 1.017(0.34) | 1.18(0.337) | 0.793(0.059) |
| 2000 | 1.215(1.235) | 0.673(0.281) | 1.251(1.168) | 2.272(2.169) | 0.648(0.263) | 1.204(0.267) | 0.877(0.095) | 1.073(0.18) | 1.217(0.216) | 0.802(0.049) |
| 5000 | 1.802(1.776) | 0.701(0.15) | 1.093(0.666) | 1.874(0.635) | 0.76(0.177) | 1.161(0.211) | 0.905(0.095) | 1.143(0.175) | 1.212(0.157) | 0.841(0.068) |
| 10000 | 1.506(0.848) | 0.713(0.122) | 1.127(0.455) | 1.722(0.475) | 0.7(0.096) | 1.268(0.314) | 0.935(0.048) | 1.22(0.176) | 1.278(0.122) | 0.874(0.05) |
| 20000 | 1.332(0.571) | 0.717(0.126) | 0.977(0.266) | 1.555(0.297) | 0.785(0.087) | 1.372(0.169) | 0.963(0.032) | 1.19(0.11) | 1.186(0.082) | 0.872(0.035) |
| 40000 | 1.113(0.076) | 0.716(0.014) | 1.055(0.038) | 1.684(0.215) | 0.806(0.005) | 1.275(0.003) | 0.973(0.002) | 1.238(0.007) | 1.099(0.051) | 0.868(0.001) |

Appendix Table D.3: Calibration of semi-supervised and supervised models in the NLST dataset. Results are not available for some smaller training sets due to insufficient events.

| PLCO | One-year death | | | Twelve-Year Death | | |
|---|---|---|---|---|---|---|
| | Self-supervised Autoencoder | Self-Supervised PCLR | Self-Supervised MoCo | Self-Supervised Autoencoder | Self-Supervised PCLR | Self-Supervised MoCo |
| 200 | N/A | N/A | N/A | 1.032(0.206) | 1.024(0.206) | 1.027(0.202) |
| 500 | N/A | N/A | N/A | 1.048(0.084) | 1.052(0.079) | 1.052(0.08) |
| 1000 | N/A | N/A | N/A | 1.006(0.069) | 1.003(0.066) | 1.003(0.067) |
| 2000 | 1.209(0.249) | 1.114(0.412) | 1.112(0.413) | 1.02(0.057) | 1.017(0.055) | 1.014(0.058) |
| 5000 | 1.135(0.119) | 1.201(0.249) | 1.202(0.247) | 1.024(0.033) | 1.021(0.033) | 1.019(0.032) |
| 10000 | 1.227(0.087) | 1.136(0.116) | 1.133(0.119) | 1.033(0.017) | 1.031(0.018) | 1.028(0.017) |
| 20000 | 1.218(0.231) | 1.221(0.086) | 1.219(0.085) | 1.035(0.016) | 1.033(0.017) | 1.031(0.015) |
| 40000 | 1.204(0.002) | 1.206(0.0) | 1.201(0.001) | 1.034(0.0) | 1.031(0.0) | 1.031(0.0) |

Appendix Table D.4: Calibration of Self-Supervised models in the PLCO dataset

| NLST | One-year death | | | Twelve-Year Death | | |
|---|---|---|---|---|---|---|
| | Self-Supervised Autoencoder | Self-Supervised PCLR | Self-Supervised MoCo | Self-Supervised Autoencoder | Self-Supervised PCLR | Self-Supervised MoCo |
| 200 | N/A | N/A | N/A | 1.361(1.167) | 0.864(0.224) | 0.778(0.17) |
| 500 | N/A | N/A | N/A | 0.831(0.196) | 0.907(0.096) | 0.806(0.062) |
| 1000 | N/A | N/A | N/A | 1.2(0.894) | 0.86(0.106) | 0.781(0.06) |
| 2000 | 0.935(0.611) | 0.673(0.281) | 0.638(0.205) | 1.104(0.68) | 0.877(0.095) | 0.794(0.062) |
| 5000 | 0.965(0.783) | 0.701(0.15) | 0.712(0.147) | 0.566(0.314) | 0.905(0.095) | 0.784(0.033) |
| 10000 | 0.958(0.592) | 0.713(0.122) | 0.675(0.07) | 0.389(0.142) | 0.935(0.048) | 0.805(0.025) |
| 20000 | 1.196(0.649) | 0.717(0.126) | 0.723(0.052) | 0.379(0.068) | 0.963(0.032) | 0.826(0.021) |
| 40000 | 1.004(0.002) | 0.716(0.014) | 0.711(0.005) | 0.388(0.006) | 0.973(0.002) | 0.831(0.001) |

Appendix Table D.5: Calibration of self-supervised models in the NLST dataset.

## Appendix E. Hyperparameter Sensitivity Analysis

We performed a sensitivity analysis on the hyperparameters for the Autoencoder-Classifier's loss function $\lambda_{reg}, \lambda_{recon}, \lambda_{norm}$. The default parameters were $(1, 20, 0.0001)$. We tested the following configurations:

- Amplified Classification & Regression parameter $\lambda_{reg}$: $(10, 20, 0.0001)$

- Amplified Image reconstruction parameter $\lambda_{recon}$: $(1, 200, 0.0001)$

- Amplified Normalization parameter $\lambda_{norm}$: $(1, 20, 0.001)$

Using each parameter setting, we trained separate models on a randomly selected set of 7000 radiographs from the pretraining datasets and tested them on PLCO and NLST to predict 12-year mortality: Figure E.9 and Table D.1.
The results for different hyperparameters are generally consistent with each other. However, amplification of the classification loss shows a slight disadvantage for internal validation when the training set size is large, and a slight advantage for both internal and external validation when the training set size is small. It is possible that a larger discrepancy may emerge if a larger subset of images is used for training. Currently, scaling up any of the hyperparameters does not significantly impact the AUC showing the robustness of the approach.

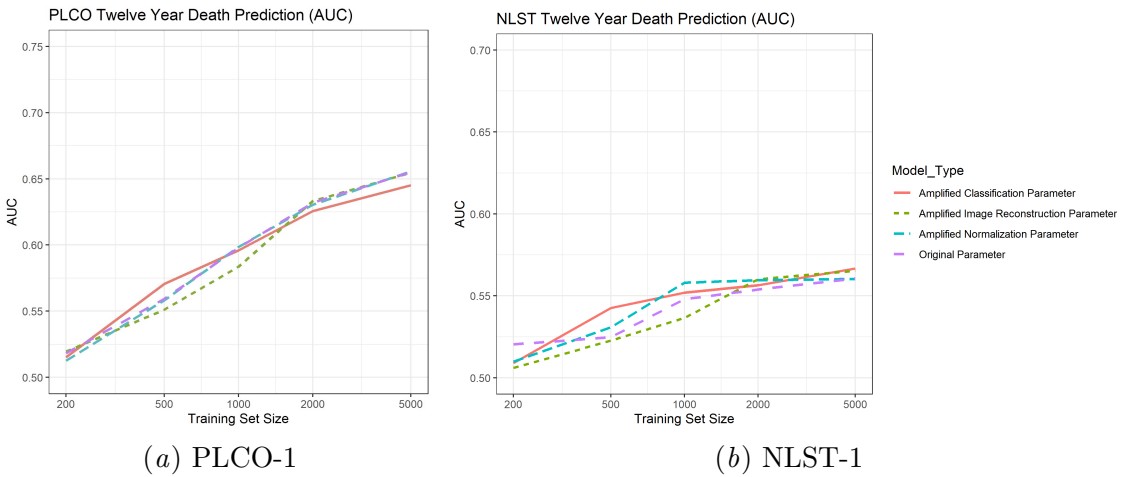

($a$) PLCO-1    ($b$) NLST-1

Appendix Figure E.9: AUC comparisons for 12-year mortality rate between different hyperparameter settings.

## Appendix F. Prediction of intermediate risk factors

| | Target Variable | PLCO Internal Validation | | | |
| --- | --- | --- | --- | --- | --- |
| | | Semi-Supervised Autoencoder | Semi-Supervised MoCo | Semi-Supervised PCLR | Transfer Learning |
| Continuous Demographics and Risk Factors | Age | 0.654 (0.64,0.66) | 0.465 (0.45,0.48) | 0.392 (0.38,0.41) | 0.373 (0.36,0.39) |
| | BMI | 0.823 (0.82,0.83) | 0.629 (0.62,0.64) | 0.732 (0.72,0.74) | 0.656 (0.65,0.67) |
| | Pack-Years | 0.332 (0.31,0.35) | 0.205 (0.18,0.23) | 0.217 (0.2,0.24) | 0.202 (0.18,0.22) |
| Discrete Demographics and Risk Factors | Black Race | 0.947 (0.94,0.95) | 0.776 (0.76,0.79) | 0.845 (0.83,0.86) | 0.794 (0.78,0.81) |
| | History of Type 2 Diabetes | 0.752 (0.74,0.77) | 0.669 (0.65,0.69) | 0.709 (0.69,0.72) | 0.667 (0.65,0.68) |
| | History of Emphysema | 0.799 (0.77,0.82) | 0.677 (0.65,0.7) | 0.676 (0.65,0.7) | 0.684 (0.66,0.71) |
| | History of Smoking | 0.675 (0.67,0.68) | 0.622 (0.61,0.63) | 0.63 (0.62,0.64) | 0.622 (0.61,0.63) |
| | History of Myocardial Infarction | 0.763 (0.75,0.78) | 0.673 (0.66,0.69) | 0.7 (0.68,0.72) | 0.662 (0.65,0.68) |
| | Hispanic Ethnicity | 0.789 (0.76,0.82) | 0.728 (0.7,0.76) | 0.746 (0.72,0.78) | 0.684 (0.65,0.72) |
| | History of Hypertension | 0.696 (0.69,0.7) | 0.644 (0.63,0.65) | 0.661 (0.65,0.67) | 0.649 (0.64,0.66) |
| | History of Osteoporosis | 0.804 (0.79,0.82) | 0.76 (0.74,0.78) | 0.745 (0.73,0.76) | 0.708 (0.69,0.73) |
| | History of Cancer | 0.649 (0.63,0.67) | 0.619 (0.6,0.64) | 0.629 (0.61,0.65) | 0.602 (0.58,0.62) |
| | Sex | 0.999 (1,1) | 0.991 (0.99,0.99) | 0.987 (0.99,0.99) | 0.958 (0.96,0.96) |
| | History of Stroke | 0.672 (0.64,0.7) | 0.602 (0.57,0.63) | 0.62 (0.59,0.65) | 0.593 (0.56,0.62) |
| Radiologist Findings | Atelectasis | 0.753 (0.56,0.95) | 0.523 (0.26,0.78) | 0.628 (0.43,0.83) | 0.576 (0.29,0.86) |
| | Bone/Chest Wall Lesion | 0.761 (0.74,0.78) | 0.694 (0.67,0.72) | 0.731 (0.71,0.75) | 0.665 (0.64,0.69) |
| | Cardiovascular Abnormality | 0.89 (0.88,0.9) | 0.786 (0.77,0.81) | 0.811 (0.79,0.83) | 0.81 (0.79,0.83) |
| | COPD/Emphysema | 0.842 (0.82,0.86) | 0.752 (0.73,0.78) | 0.785 (0.76,0.81) | 0.745 (0.72,0.77) |
| | Lung Fibrosis | 0.704 (0.69,0.72) | 0.608 (0.59,0.62) | 0.612 (0.6,0.63) | 0.629 (0.61,0.65) |
| | Lung Opacity | 0.68 (0.63,0.73) | 0.569 (0.52,0.62) | 0.562 (0.51,0.61) | 0.586 (0.53,0.64) |
| | Lymphadenopathy | 0.725 (0.67,0.78) | 0.67 (0.62,0.72) | 0.59 (0.52,0.66) | 0.563 (0.5,0.63) |
| | Lung Nodule | 0.617 (0.6,0.63) | 0.59 (0.57,0.61) | 0.593 (0.57,0.61) | 0.559 (0.54,0.58) |
| | Pleural Fibrosis | 0.723 (0.7,0.75) | 0.646 (0.62,0.67) | 0.633 (0.61,0.66) | 0.613 (0.59,0.64) |

Appendix Table F.6: Comparison of prediction of intermediate risk factors for supervised and semi-supervised models on the PLCO dataset using Pearson Correlation for continuous variables and AUC for discrete variables. Results indicate superior performance of the semi-supervised autoencoder in most target tasks.

| | | NLST External Validation | | | |
|---|---|---|---|---|---|
| | Target Variable | Semi-Supervised Autoencoder | Semi-Supervised MoCo | Semi-Supervised PCLR | Transfer Learning |
| Continuous Demographics and Risk Factors | Age | 0.547 (0.53,0.57) | 0.465 (0.44,0.49) | 0.226 (0.2,0.25) | 0.407 (0.38,0.43) |
| | BMI | 0.761 (0.75,0.77) | 0.493 (0.47,0.51) | 0.63 (0.61,0.65) | 0.68 (0.67,0.69) |
| | Pack-Years | 0.181 (0.16,0.21) | 0.166 (0.14,0.19) | 0.101 (0.07,0.13) | 0.156 (0.13,0.18) |
| Discrete Demographics and Risk Factors | History of Type 2 Diabetes | 0.737 (0.72,0.76) | 0.651 (0.63,0.68) | 0.679 (0.66,0.7) | 0.704 (0.68,0.73) |
| | History of Emphysema | 0.609 (0.58,0.64) | 0.574 (0.54,0.6) | 0.573 (0.54,0.6) | 0.637 (0.61,0.67) |
| | History of Myocardial Infarction | 0.669 (0.65,0.69) | 0.625 (0.6,0.65) | 0.622 (0.6,0.64) | 0.657 (0.63,0.68) |
| | History of Hypertension | 0.634 (0.62,0.65) | 0.595 (0.58,0.61) | 0.608 (0.59,0.62) | 0.644 (0.63,0.66) |
| | History of Cancer | 0.637 (0.6,0.68) | 0.602 (0.56,0.64) | 0.543 (0.51,0.58) | 0.576 (0.54,0.61) |
| | Sex | 0.993 (0.99,1) | 0.974 (0.97,0.98) | 0.905 (0.9,0.91) | 0.917 (0.91,0.92) |
| | History of Stroke | 0.596 (0.55,0.64) | 0.528 (0.48,0.57) | 0.578 (0.53,0.62) | 0.601 (0.56,0.64) |
| Radiologist Findings | Atelectasis | 0.611 (0.53,0.69) | 0.501 (0.41,0.59) | 0.566 (0.47,0.66) | 0.588 (0.5,0.67) |
| | Bone/Chest Wall Lesion | 0.608 (0.54,0.68) | 0.511 (0.43,0.59) | 0.569 (0.49,0.64) | 0.507 (0.43,0.58) |
| | Cardiovascular Abnormality | 0.721 (0.67,0.77) | 0.77 (0.73,0.81) | 0.715 (0.67,0.76) | 0.808 (0.76,0.85) |
| | COPD/Emphysema | 0.638 (0.62,0.66) | 0.6 (0.58,0.62) | 0.643 (0.63,0.66) | 0.679 (0.66,0.69) |
| | Lung Fibrosis | 0.606 (0.58,0.63) | 0.572 (0.55,0.59) | 0.51 (0.49,0.53) | 0.542 (0.52,0.56) |
| | Lung Opacity | 0.65 (0.55,0.75) | 0.539 (0.43,0.65) | 0.496 (0.4,0.59) | 0.534 (0.41,0.65) |
| | Lymphadenopathy | 0.615 (0.53,0.7) | 0.521 (0.44,0.6) | 0.615 (0.54,0.69) | 0.526 (0.44,0.61) |
| | Lung Nodule | 0.544 (0.53,0.56) | 0.521 (0.5,0.54) | 0.527 (0.51,0.54) | 0.541 (0.52,0.56) |
| | Pleural Fibrosis | 0.514 (0.49,0.54) | 0.558 (0.53,0.58) | 0.499 (0.47,0.52) | 0.598 (0.57,0.62) |

Appendix Table F.7: Prediction accuracy of intermediate risk factors using supervised and semi-supervised models on the NLST external validation dataset. We observed similar findings as in the internal validation (PLCO) results.

| | | PLCO Internal Validation | | | NLST External Validation | | |
|---|---|---|---|---|---|---|---|
| | Target Variable | Self-Supervised PCLR | Self-Supervised Autoencoder | Self-Supervised MoCo | Self-Supervised PCLR | Self-Supervised Autoencoder | Self-Supervised MoCo |
| Continuous Demographics and Risk Factors | Age | 0.469 (0.46,0.48) | 0.433 (0.42,0.45) | 0.196 (0.18,0.21) | 0.368 (0.34,0.39) | 0.126 (0.1,0.15) | 0.121 (0.09,0.15) |
| | BMI | 0.744 (0.74,0.75) | 0.73 (0.72,0.74) | 0.509 (0.5,0.52) | 0.692 (0.68,0.71) | 0.464 (0.44,0.48) | 0.349 (0.33,0.37) |
| | Pack-Years | 0.246 (0.22,0.27) | 0.226 (0.21,0.25) | 0.171 (0.15,0.19) | 0.153 (0.13,0.18) | 0.044 (0.02,0.07) | 0.083 (0.06,0.11) |
| Discrete Demographics and Risk Factors | History of Type 2 Diabetes | 0.715 (0.7,0.73) | 0.689 (0.67,0.7) | 0.635 (0.62,0.65) | 0.705 (0.68,0.73) | 0.558 (0.53,0.59) | 0.58 (0.55,0.61) |
| | History of Emphysema | 0.709 (0.68,0.74) | 0.672 (0.64,0.7) | 0.64 (0.61,0.67) | 0.617 (0.59,0.65) | 0.542 (0.51,0.57) | 0.567 (0.54,0.6) |
| | History of Smoking | 0.641 (0.63,0.65) | 0.629 (0.62,0.64) | 0.615 (0.61,0.62) | NA | NA | NA |
| | History of Myocardial Infarction | 0.712 (0.7,0.73) | 0.689 (0.67,0.7) | 0.631 (0.62,0.65) | 0.652 (0.63,0.67) | 0.546 (0.52,0.57) | 0.571 (0.55,0.59) |
| | History of Hypertension | 0.675 (0.67,0.68) | 0.667 (0.66,0.68) | 0.606 (0.6,0.62) | 0.629 (0.61,0.64) | 0.559 (0.54,0.57) | 0.564 (0.55,0.58) |
| | History of Osteoporosis | 0.762 (0.75,0.78) | 0.747 (0.73,0.76) | 0.721 (0.7,0.74) | NA | NA | NA |
| | History of Cancer | 0.633 (0.61,0.66) | 0.616 (0.59,0.64) | 0.631 (0.61,0.65) | 0.563 (0.52,0.6) | 0.537 (0.5,0.58) | 0.503 (0.47,0.54) |
| | Sex | 0.992 (0.99,0.99) | 0.982 (0.98,0.98) | 0.943 (0.94,0.95) | 0.949 (0.94,0.95) | 0.754 (0.74,0.77) | 0.728 (0.71,0.74) |
| | History of Stroke | 0.626 (0.59,0.66) | 0.585 (0.55,0.62) | 0.532 (0.5,0.57) | 0.551 (0.51,0.59) | 0.504 (0.46,0.55) | 0.511 (0.47,0.56) |
| Radiologist Findings | Atelectasis | 0.556 (0.29,0.82) | 0.613 (0.47,0.76) | 0.54 (0.4,0.68) | 0.548 (0.46,0.63) | 0.548 (0.47,0.63) | 0.551 (0.46,0.64) |
| | Bone/Chest Wall Lesion | 0.727 (0.7,0.75) | 0.704 (0.68,0.73) | 0.689 (0.67,0.71) | 0.569 (0.5,0.64) | 0.615 (0.54,0.69) | 0.499 (0.42,0.57) |
| | Cardiovascular Abnormality | 0.841 (0.82,0.86) | 0.79 (0.77,0.81) | 0.736 (0.71,0.76) | 0.809 (0.77,0.85) | 0.628 (0.57,0.68) | 0.639 (0.59,0.69) |
| | COPD/Emphysema | 0.783 (0.76,0.81) | 0.775 (0.75,0.8) | 0.697 (0.67,0.72) | 0.65 (0.63,0.67) | 0.597 (0.58,0.61) | 0.567 (0.55,0.59) |
| | Lung Fibrosis | 0.647 (0.63,0.66) | 0.603 (0.59,0.62) | 0.564 (0.55,0.58) | 0.6 (0.58,0.62) | 0.503 (0.48,0.53) | 0.512 (0.49,0.54) |
| | Lung Opacity | 0.607 (0.55,0.66) | 0.541 (0.49,0.59) | 0.556 (0.5,0.61) | 0.533 (0.45,0.62) | 0.538 (0.45,0.63) | 0.497 (0.42,0.58) |
| | Lymphadenopathy | 0.671 (0.61,0.73) | 0.634 (0.58,0.69) | 0.58 (0.52,0.64) | 0.566 (0.48,0.65) | 0.505 (0.41,0.6) | 0.565 (0.49,0.64) |
| | Lung Nodule | 0.59 (0.57,0.61) | 0.576 (0.56,0.59) | 0.572 (0.55,0.59) | 0.542 (0.52,0.56) | 0.524 (0.51,0.54) | 0.513 (0.5,0.53) |
| | Pleural Fibrosis | 0.66 (0.63,0.69) | 0.624 (0.6,0.65) | 0.607 (0.58,0.63) | 0.58 (0.55,0.61) | 0.509 (0.48,0.53) | 0.519 (0.49,0.55) |

Appendix Table F.8: Prediction accuracy of intermediate risk factors using self-supervised models on PLCO (internal validation) and NLST (external validation). Results indicate superior performance of Self-Supervised Contrastive Learning and Autoencoder in most tasks.

## Appendix G.  A Comparison of Results of Different Downstream Task Models

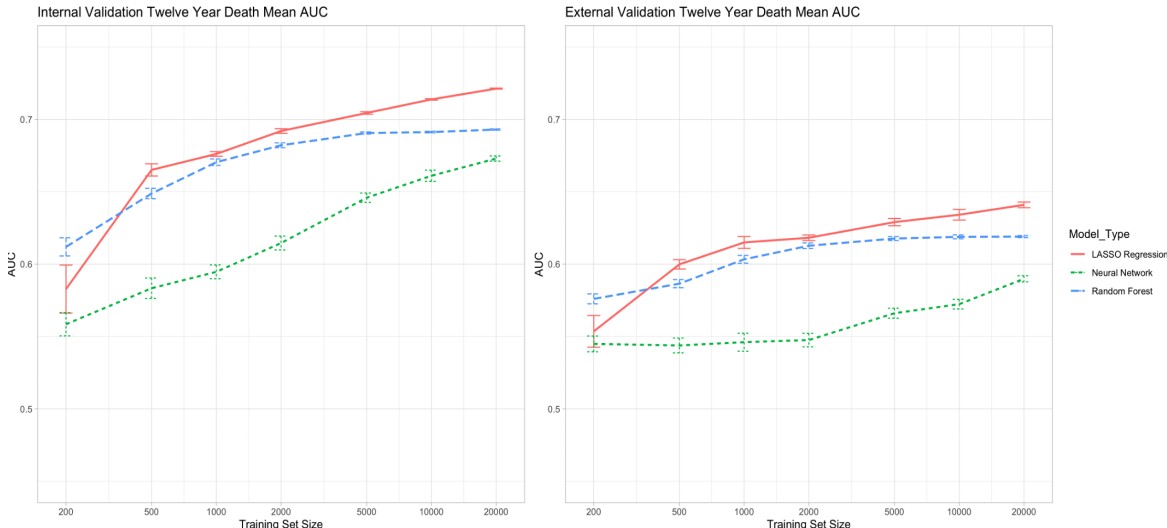

Appendix Figure G.10: The performance of LASSO regression, a neural network with one hidden layer and nonlinear activation, and a random forest with 10 max nodes and 10 trees were evaluated using AUC on 12-year mortality rate prediction with the PLCO dataset. Results showed that LASSO had the best AUC performance across all training sizes. However, the neural network failed to converge in most trials of size 40,000, therefore trial size 40,000 was not included in the final analysis.

# Appendix H. Effect of cohort differences on validation set performance

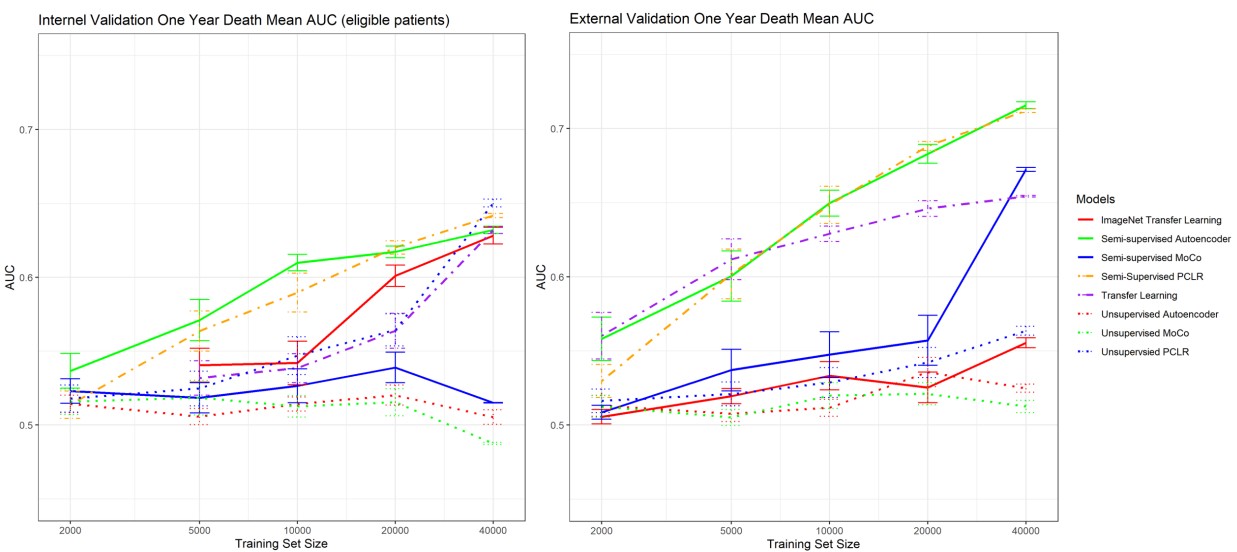

Appendix Figure H.11: Left: PLCO internal validation results (only participants meeting NLST entry criteria), Right: NLST external validation results.

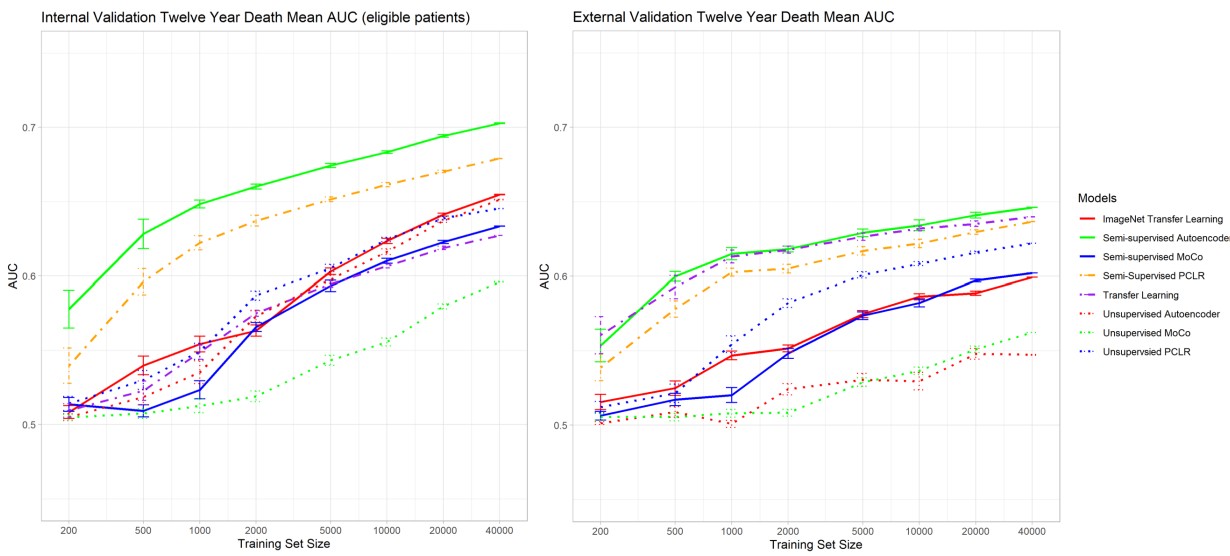

Appendix Figure H.12: Left: PLCO internal validation results (only participants meeting NLST entry criteria), Right: NLST external validation results.

# Appendix I.  Cohort Characteristics

| | Pretraining | | Training | Testing | |
|---|---|---|---|---|---|
| | CheXpert | NIH-CXR14 | PLCO Training | PLCO Internal Testing | NLST External Testing |
| Description | All adults with posterior-anterior chest x-rays from Stanford University Hospital between October 2002 and July 2017. | All adults with posterior-anterior chest x-rays from the NIH Clinical Center between 1992 and 2015. | Adults in the chest x-ray arm of the Prostate, Lung, Colorectal, and Ovarian cancer screening trial. Adults 55-74 years from 21 US sites. | Testing set individuals from PLCO trial. | Adults in the chest x-ray arm of the National Lung Screening Trial. Heavy cigarette smokers 55-74 years with no history of lung cancer from 10 US sites |
| Demographics | | | | | |
| Age | 57.0 (17.2) | 48.8 (14.8) | 62.4 (5.4) | 62.4 (5.4) | 61.7 (5.0) |
| Female Sex | 10134 / 29419 (34.4%) | 28965 / 64628 (44.8%) | 18073 / 37467 (48.2%) | 6631 / 13723 (48.3%) | 2416 / 5414 (44.6%) |
| Race | | | | | |
| White | NA | NA | 32524 / 37433 (86.7%) | 11915 / 13708 (86.9%) | 5088 / 5389 (55.9%) |
| Black | NA | NA | 2194 / 37433 (5.9%) | 807 / 13708 (5.9%) | 208 / 5389 (3.9%) |
| Other | NA | NA | 2010 / 37433 (5.4%) | 732 / 13708 (5.3%) | 93 / 5389 (1.7%) |
| Radiology Findings | | | | | |
| No Finding | 9589 / 29419 (32.6%) | 37452 / 64628 (58.0%) | NA | NA | NA |
| Pleural Effusion | 8078 / 29419 (27.5%) | 6450 / 64628 (10.0%) | NA | NA | NA |
| Lung Opacity | 9736 / 29419 (33.1%) | NA | 292 / 37467 (0.8%) | 111 / 13723 (0.8%) | 33 / 5414 (0.6%) |
| Lung Lesion | 2122 / 29419 (7.2%) | 6972 / 64628 (10.8%) | 2892 / 37467 (7.7%) | 971 / 13723 (7.1%) | 1610 / 5414 (29.7%) |
| Infiltration | NA | 8976 / 64628 (13.9%) | NA | NA | NA |
| Atelectasis | 3195 / 29419 (10.9%) | 5614 / 64628 (8.7%) | 20 / 37467 (0.0%) | 6 / 13723 (0.0%) | 43 / 5414 (0.8%) |
| Pneumothorax | 1802 / 29419 (6.1%) | 3268 / 64628 (5.1%) | NA | NA | NA |
| Cardiac Abnormality | NA | NA | 1496 / 37467 (4.0%) | 536 / 13723 (3.9%) | 122 / 5414 (2.3%) |
| Cardiomegaly | 2909 / 29419 (9.9%) | 1520 / 64628 (2.4%) | NA | NA | NA |
| Consolidation | 1498 / 29419 (5.1%) | 1463 / 64628 (2.3%) | NA | NA | NA |
| Pleural Thickening | NA | 2366 / 64628 (3.7%) | NA | NA | NA |
| Edema | 1709 / 29419 (5.8%) | 268 / 64628 (0.4%) | NA | NA | NA |
| Pneumonia | 1198 / 29419 (4.1%) | 586 / 64628 (0.9%) | NA | NA | NA |
| Emphysema | NA | 1473 / 64628 (2.3%) | 974 / 37467 (2.6%) | 371 / 13723 (2.7%) | 1197 / 5414 (22.1%) |
| Enlarged Cardiomediastinum | 1436 / 29419 (4.9%) | NA | NA | NA | |
| Fracture | 1411 / 29419 (4.8%) | NA | NA | NA | |
| Lung Fibrosis | NA | 1394 / 64628 (2.2%) | 2858 / 37467 (7.6%) | 1059 / 13723 (7.7%) | 700 / 5414 (12.9%) |
| Other Pleural Findings | 1012 / 29419 (3.4%) | NA | 1184 / 37467 (3.2%) | 416 / 13723 (3.0%) | 471 / 5414 (8.7%) |
| Hernia | NA | 191 / 64628 (0.3%) | NA | NA | NA |
| Bone/Chest Wall Lesion | NA | NA | 1663 / 37467 (4.4%) | 596 / 13723 (4.3%) | 61 / 5414 (1.1%) |
| Lymphadenopathy | NA | NA | 230 / 37467 (0.6%) | 71 / 13723 (0.5%) | 46 / 5414 (0.8%) |
| Risk Factors | | | | | |
| BMI | NA | NA | 27.3 (4.9) | 27.3 (4.9) | 27.8 (5.3) |
| Pack-Years | NA | NA | 35.2 (28.8) | 35.6 (29.6) | 55.7 (23.5) |
| History of Type 2 Diabetes | NA | NA | 2864 / 37410 (7.7%) | 1046 / 13694 (7.6%) | 501 / 5402 (9.3%) |
| History of Emphysema | NA | NA | 861 / 37396 (2.3%) | 335 / 13697 (2.4%) | 642 / 5414 (11.9%) |
| History of MI | NA | NA | 3255 / 37405 (8.7%) | 1207 / 13692 (8.8%) | 665 / 5391 (12.3%) |
| History of Hypertension | NA | NA | 12432 / 37414 (33.2%) | 4633 / 13700 (33.8%) | 1985 / 5399 (36.8%) |
| History of Cancer | NA | NA | 1632 / 37453 (4.4%) | 576 / 13716 (4.2%) | 225 / 5414 (4.2%) |
| History of Stroke | NA | NA | 845 / 37407 (2.3%) | 308 / 13697 (2.2%) | 173 / 5392 (3.2%) |
| Outcomes | | | | | |
| 1-year mortality | NA | NA | 153 / 37467 (0.5%) | 48 / 13723 (0.3%) | 33 / 5414 (0.6%) |
| 12-year mortality | NA | NA | 4983 / 37467 (13.3%) | 1800 / 13723 (13.1%) | 931 / 5414 (17.2%) |

Appendix Table I.9:  Cohort Characteristics

## Appendix J. Effect of random weight initialization

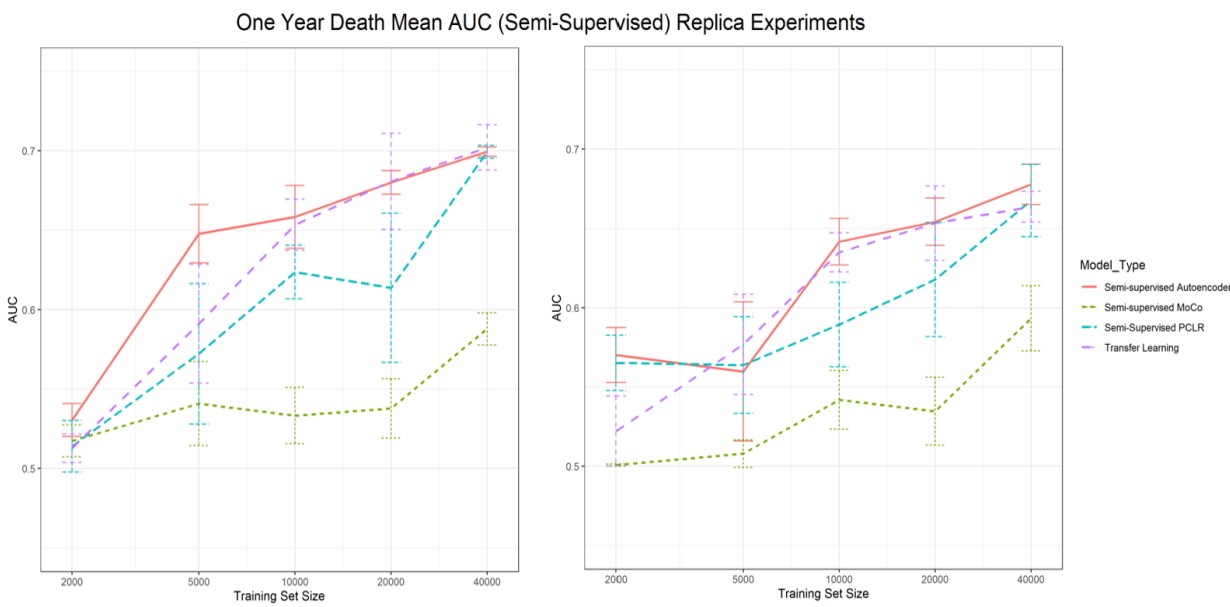

Appendix Figure J.13: Left: internal validation (PLCO), Right: external validation (NLST).

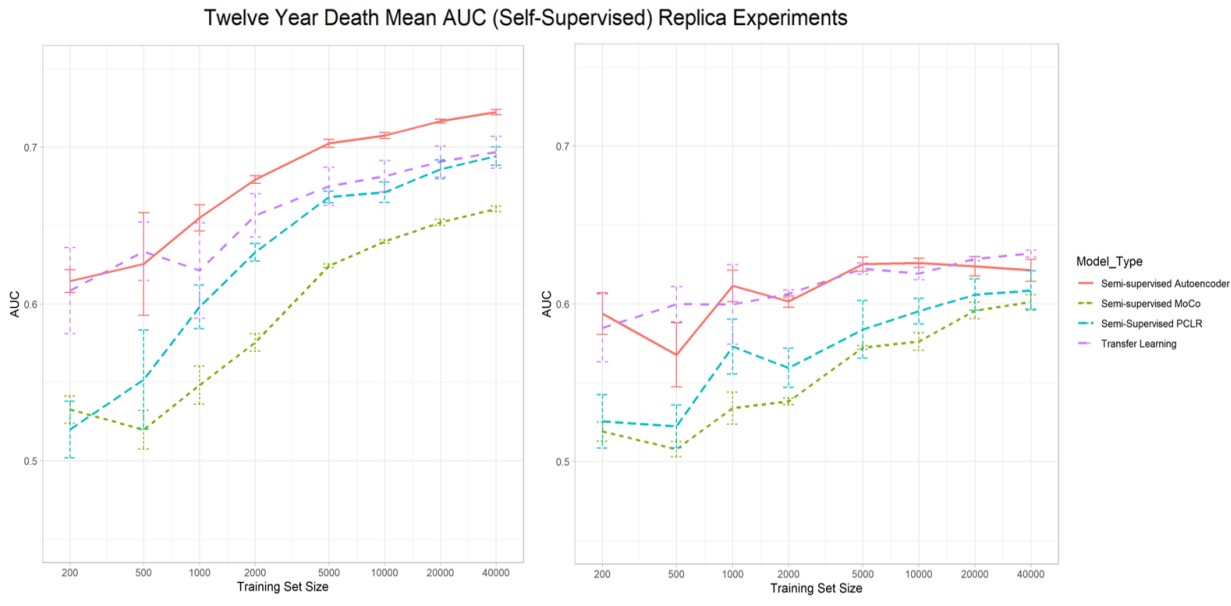

Appendix Figure J.14: Left: internal validation (PLCO), Right: external validation (NLST).

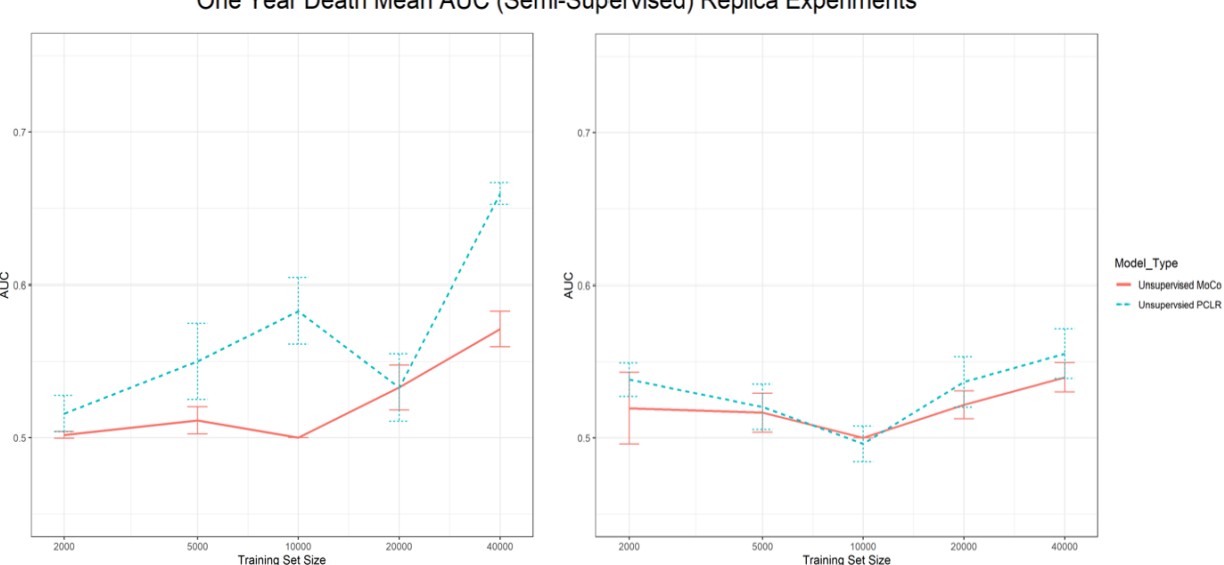

Appendix Figure J.15: Left: internal validation (PLCO), Right: external validation (NLST).

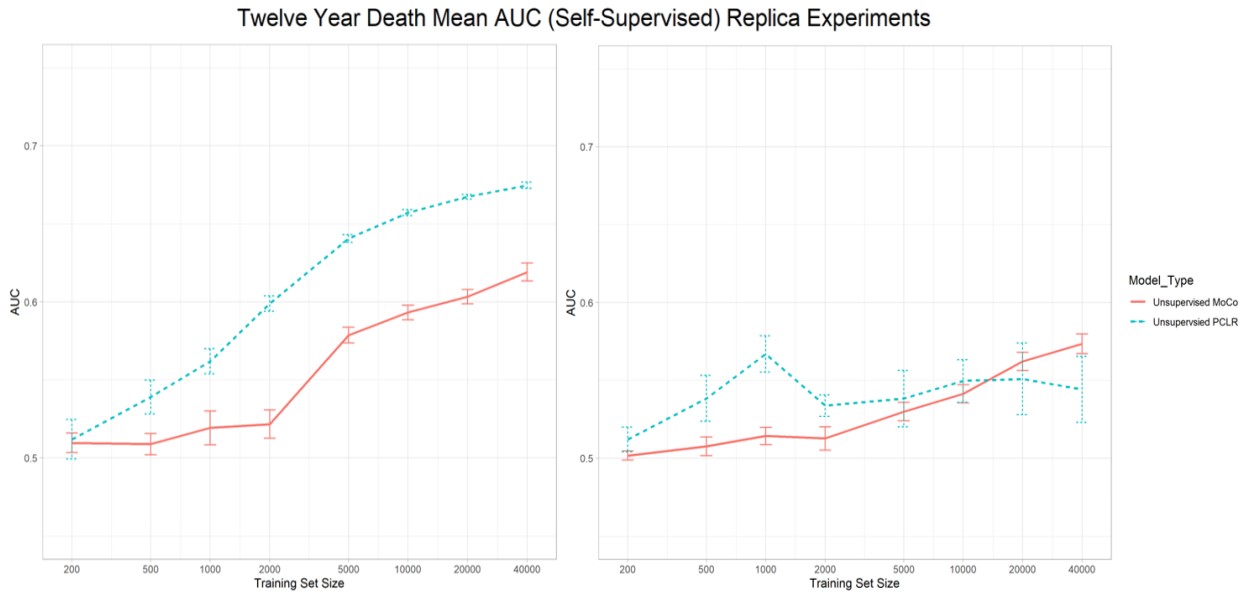

Appendix Figure J.16: Left: internal validation (PLCO), Right: external validation (NLST).

## Appendix K. Effect of Image resolution

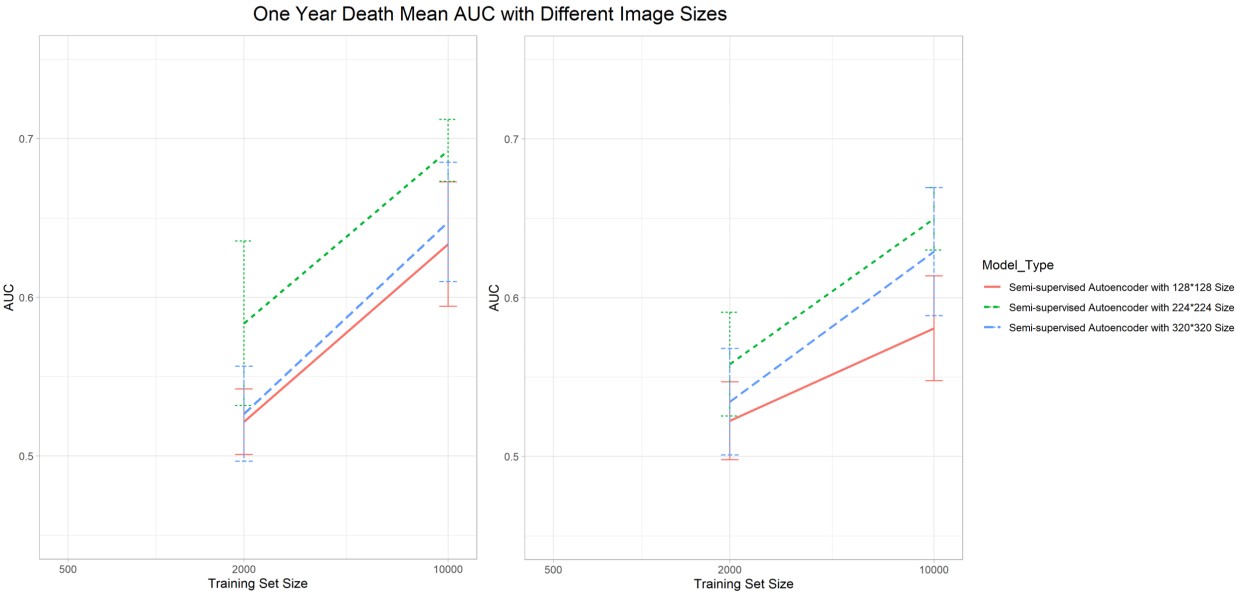

Appendix Figure K.17: Left: internal validation (PLCO), Right: external validation (NLST). Pre-training models were trained using 128*128, 224*224 (original semi-supervised autoencoder model), and 320*320 as pre-training image sizes. All other procedures, including encodings extraction and experiments, are held the same.

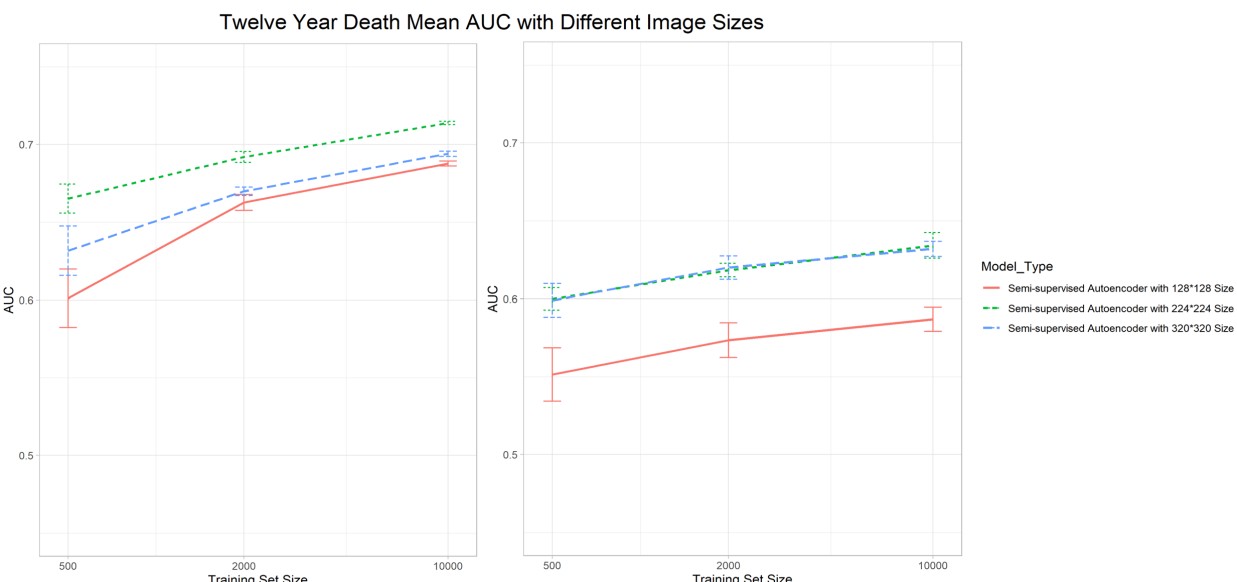

Appendix Figure K.18: Left: internal validation (PLCO), Right: external validation (NLST). Pre-training models were trained using 128*128, 224*224 (original semi-supervised autoencoder model), and 320*320 as pre-training image sizes. All other procedures, including encodings extraction and experiments, are held the same.

## Appendix L. Autoencoder Ablation Experiments

Here, we show an ablation experiment to show the impact of each component of the semi-supervised autoencoder model. Our major finding is that semi-supervision contributes the most to performance; however, the self-supervision adds additional value for rare outcomes (1-year mortality).

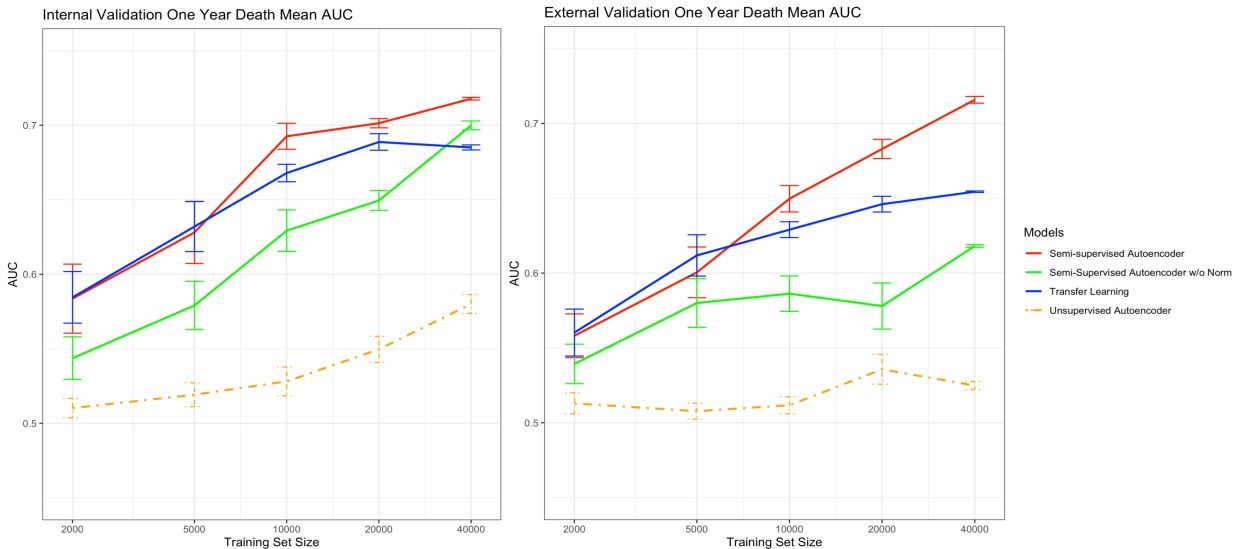

Appendix Figure L.19: Left: PLCO internal validation results, Right: NLST external validation results.

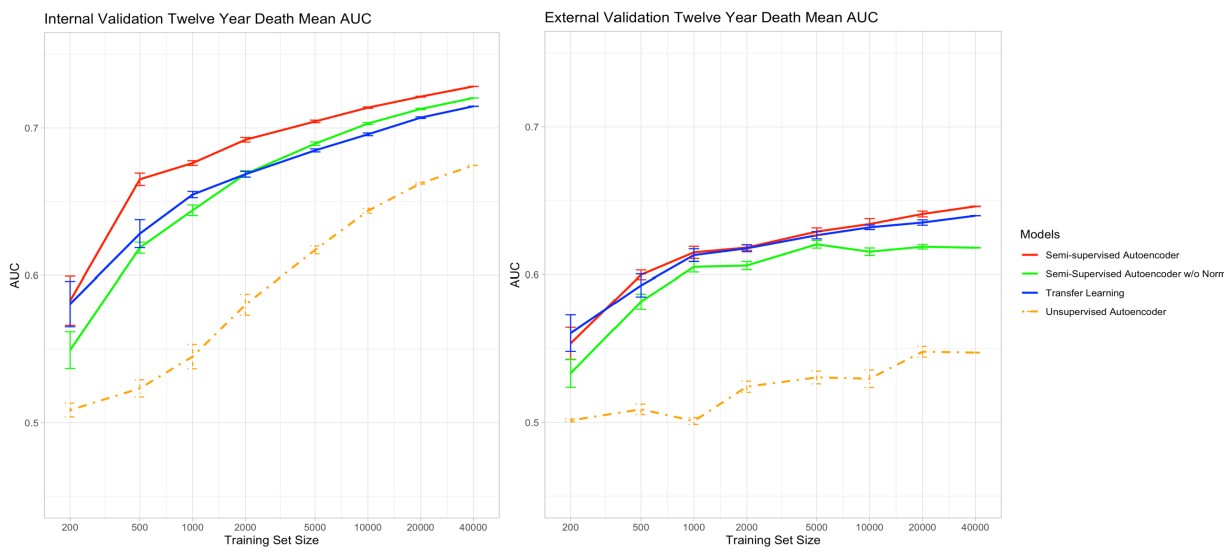

Appendix Figure L.20: Left: PLCO internal validation results, Right: NLST external validation results.

