# OpenReview forum: "A comparison of self-supervised pretraining approaches for predicting disease risk from chest radiograph images"
_MIDL.io/2023/Conference — MIDL 2023 Poster_

### Official Review · Reviewer_BXfD · 2023-02-06

**Confidence:** 4
**Preliminary Rating:** 3
**Recommendation:** Oral

**Summary:**

The aim of this study is to evaluate self- and semi-supervised pretraining techniques for predicting mortality risk from CXRs considering a novel semi-supervised autoencoder for representation learning, pretraining strategies using target variables with varying class imbalance, the generalizability of representations in an external validation dataset, and predicting disease history and risk factors. This is quite interesting topics. The study is well designed, and there are well descriptions on details on how to preprocessing, training, and validation. However, 224x224 is too small for survival predicition using CXR, because this size will be missing most of nodules. In addition, there is a lack of technical originality.

**Strengths:**

The aim of this study is to evaluate self- and semi-supervised pretraining techniques for predicting mortality risk from CXRs considering a novel semi-supervised autoencoder for representation learning, pretraining strategies using target variables with varying class imbalance, the generalizability of representations in an external validation dataset, and predicting disease history and risk factors.

**Weaknesses:**

The study is well designed, and there are well descriptions on details on how to preprocessing, training, and validation. However, 224x224 is too small for survival predicition using CXR, because this size will be missing most of nodules.  In addition, there is a lack of technical originality.

**Deanonymize Review:**

yes

**Paper Type:**

validation/application paper

**Questions To Address In The Rebuttal:**

The study is well designed, and there are well descriptions on details on how to preprocessing, training, and validation. However, 224x224 is too small for survival predicition using CXR, because this size will be missing most of nodules.  In addition, there is a lack of technical originality.

---

### Official Review · Reviewer_pFqM · 2023-02-06

**Confidence:** 4
**Preliminary Rating:** 4
**Recommendation:** Poster

**Summary:**

A comparison of self-supervised and semi-supervised pretraining is presented for mortality risk prediction in chest X-rays. A large pretraining dataset is available with off-target annotations. semi-supervised pretraining is performed using these off-target labels. the different approaches are compared to training from scratch and pretraining on imagenet. the features acquired from the pretraining stage are subsequently used in a lasso regression to predict all-cause mortality using a training dataset.  Experiments are performed  for different training set sizes and the performance is evaluated in an internal testing dataset and an external testing dataset.

**Strengths:**

Clearly written paper.  Quite an extensive comparison of different pre-training strategies is presented. A large amount of data is used from different sites and different populations. Moreover, a clinically relevant application is addressed.

**Weaknesses:**

The discussion is short and it is not entirely clear which results support which conclusions. In my opinion the paper would benefit from a more in depth discussion. The limitations of this study are not discussed. Moreover, the external validation is from a quite different population than the population the models were calibrated on. the paper would benefit from a more extensive description of the data and a discussion on the performance drifferences between the internal and external test set.

**Deanonymize Review:**

no

**Detailed Comments:**


The authors mention 15 trials for each training set size. Do they mean they trained 15 models with 15 different random seeds? Does this concern both the pretraining and the lasso regression?

Test sets of different sizes are used. Please also report the number of positives per dataset size.

The legend in figure 3 and 4 seems to miss the dashed lines (unsupervised?)

If the authors need space for additional descriptions: in my opinion the formulas for MSE and crossentropy (1 and 2) are not needed.

the terms self-supervised and un-supervised are used both. It would read better if the authors used one consistently.
Please discus the limitations of this study



**Paper Type:**

validation/application paper

**Questions To Address In The Rebuttal:**

From the description of the data it is not immediately clear which test sets are used. Please clarify. Moreover, several different datasets are used for pretraining and testing. It would be helpful to provide a bit more insight onto the differences between those datasets and the available labels.

The population of PLCO is quite different from the NLST population. This is reflected in the results, where the performance of the models calibrated for PLCO is generally lower in the NLST population. Also, the models are less well calibrated for the NLST population. The paper would benefit from a discussion on the results and the relevance. Moreover, more information on the differences between the populations is needed to understand the results. For instance, what is the incidence rate or number of positives in both populations.

---

### Official Review · Reviewer_R4WY · 2023-02-07

**Confidence:** 4
**Preliminary Rating:** 4
**Recommendation:** Poster

**Summary:**

The authors presented comparisons between self-supervised and semi-supervised autoencoder, MoCo, PCLR models, and they proposed a semi-supervised autoencoder outperforms contrastive and transfer learning in internal and external validation. They also compared with transfer learning and training from scratch, and found that a semi-supervised autoencoder and transfer learning representations were strongly associated with risk factors etc.

**Strengths:**

The reviewer appreciates authors' comprehensive comparisons between self-supervised and semi-supervised models.

The reviewer likes the adequate literature reviews.

There're plenty of details in data preprocessing and hyperparameters sensitivity analysis.

**Weaknesses:**

The authors addressed a novel semi-supervised autoencoder in their contributions, however in their experiments are only comparisons with MoCo and PCLR models.

Lack of introduction of self-supervised autoencoder architecture, meanwhile it has been outscored by self-supervised PCLR in all four settings.

The reviewer believes the major experiment is between self-supervised and semi-supervised models, however the title only referred to self-supervised approach.

There no detailed information about the semi-supervised dataset setting.

The figures' representations need improvements, e.g. the legend of fig3&4 don't match to the body.



**Deanonymize Review:**

no

**Detailed Comments:**

The authors may also consider to convey a ablation experiment which explains each layers function in the autoencoder. They may be interested in outscoring self-supervised PCLR model.

The reviewer is more likely to consider this paper as a comprehensive validation paper.

**Paper Type:**

validation/application paper

**Questions To Address In The Rebuttal:**

The reviewer believes that an additional experiment between the proposed autoencoder and baselines is necessary, which would introduce model novelty.

The authors should reconsider their title and the main part of this paper. They may either focus on model comparisons or autoencoder innovation.

---

### Meta-Review · Area_Chair_xtKJ · 2023-02-23

**Recommendation:** Accept (Poster)
**Confidence:** 5

**Metareview:**

The paper makes a comparative analysis of different semi- and self-supervised learning to predict mortality risk using chest x-ray images. While the method novelty is limited, the paper looks at a clinically relevant application and explores many strategies that may be used in practice. The paper will be of interest to the MIDL audience. However the authors need to polish the discussion an clearly correlate the conclusions with the reported findings.